# 3D Neural Scene Representations for Visuomotor Control

**Yunzhu Li**[*]
MIT CSAIL
liyunzhu@mit.edu

**Shuang Li**[*]
MIT CSAIL
lishuang@mit.edu

**Vincent Sitzmann**
MIT CSAIL
sitzmann@mit.edu

**Pulkit Agrawal**
MIT CSAIL
pulkitag@mit.edu

**Antonio Torralba**
MIT CSAIL
torralba@mit.edu

**Abstract:**
Humans have a strong intuitive understanding of the 3D environment around us. The mental model of the physics in our brain applies to objects of different materials and enables us to perform a wide range of manipulation tasks that are far beyond the reach of current robots. In this work, we desire to learn models for dynamic 3D scenes purely from 2D visual observations. Our model combines Neural Radiance Fields (NeRF) and time contrastive learning with an autoencoding framework, which learns viewpoint-invariant 3D-aware scene representations. We show that a dynamics model, constructed over the learned representation space, enables visuomotor control for challenging manipulation tasks involving both rigid bodies and fluids, where the target is specified in a viewpoint different from what the robot operates on. When coupled with an auto-decoding framework, it can even support goal specification from camera viewpoints that are *outside the training distribution*. We further demonstrate the richness of the learned 3D dynamics model by performing future prediction and novel view synthesis. Finally, we provide detailed ablation studies regarding different system designs and qualitative analysis of the learned representations.

## 1 Introduction

Existing state-of-the-art model-based systems operating from vision typically treat images as 2D grids of pixels [1, 2, 3]. The world, however, is three-dimensional. Modeling the environment from 3D enables amodal completion and allows the agents to operate from different views. Therefore, it is desirable to obtain good 3D-aware representations of the environment from 2D observations to achieve better task performance when an accurate inference of 3D information is essential, which can further make it easier to specify tasks, learn from third-person videos, etc.

One of the core questions of model learning in robotic manipulation is how to determine the state representation for learning the dynamics model. The desired representation should make it easy to capture the environment dynamics, exhibit a good 3D understanding of the objects in the scene, and be applicable to diverse object sets such as rigid or deformable objects and fluids. One line of prior work learns the dynamics model directly in the image pixel space [4, 5, 6, 7]. However, modeling dynamics in such a high-dimensional space is challenging, and these methods typically generate blurry images when performing the long-horizon future predictions. Another line of work focused on only predicting task-relevant features identified as keypoints [8, 9, 10, 11, 12]. Such models perform well in terms of category-level generalization, i.e., the same set of keypoints can represent different instances within the same category, but are not sufficient to model objects with large variations like fluids and granular materials. Other methods learn dynamics in the latent space [13, 14, 15, 2, 3]. However, the majority of these methods learn dynamics models using 2D convolutional neural networks and reconstruction loss – which has the same problem as predicting dynamics in the image

---

[*]equal contribution. Project Page: https://3d-representation-learning.github.io/nerf-dy/

5th Conference on Robot Learning (CoRL 2021), London, UK.

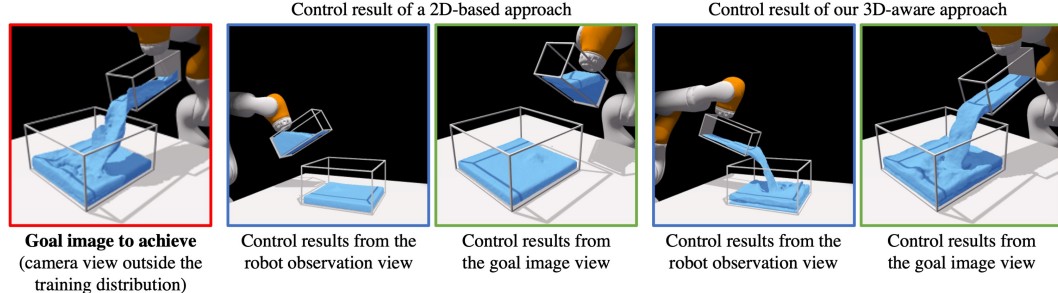

Figure 1: **Comparison of the control results between a 2D-based baseline and our 3D-aware approach.** The task here is to achieve the configuration shown on the left, observed from a viewpoint that is outside the training distribution. The agent only takes a single-view visual observation as input (images with blue frames) from a viewpoint that is vastly different from the goal. Our method generalizes well in this scenario and outperforms the 2D-based baseline, demonstrating the benefits of the learned 3D-aware scene representations.

space, i.e., their learned representations lack equivariance to 3D transformations. Time contrastive networks [16], on the other hand, aim to learn viewpoint-invariant representations from multi-view inputs, but do not require detailed modeling of 3D contents. As a result, previously unseen scene configurations and camera poses are out-of-distribution samples for the state estimator. As we will see, this leads to wrong state estimates and results in faulty control trajectories.

Meanwhile, recent work in computer vision has made impressive progress in the learning of 3D-structured neural scene representations. These approaches allow inference of 3D structure and appearance, trained only given 2D observations, either by overfitting on a single scene [17, 18, 19] or by generalizing across scenes [20, 21, 22]. Through their 3D inductive bias, the scene representations inferred by these models encode the scene contents with better accuracy and are invariant to changes in camera perspectives. It is desirable to push these ideas further to obtain a deeper understanding of how these methods, which directly reason over 3D, can bring in new characteristics and how they can be beneficial for dynamics modeling and complicated control tasks.

In this work, we aim to leverage recently proposed 3D-structure-aware implicit neural scene representations for visuomotor control tasks. We thus propose to embed neural radiance fields [19] in an auto-encoder framework, enabling tractable inference of the 3D-structure-aware scene state for dynamic environments. By additionally enforcing a time contrastive loss on the estimated states, we ensure that the learned state representations are viewpoint-invariant. We then train a dynamics model that predicts the evolution of the state space conditioned on the input action, enabling us to perform control in the learned state space. Though the representation itself is grounded in the 3D implicit field, the convolutional encoder is not. At test time, we overcome this limitation by performing inference-via-optimization [23, 20], enabling accurate state estimation even for out-of-distribution camera poses and, therefore, control of tasks where the goal view is specified in an entirely unseen camera perspective. These contributions enable us to perform model-based visuomotor control of complex scenes, modeling 3D dynamics of both rigid objects and fluids. Through comparison with various baselines, the learned representation from our model is more precise at describing the contents of 3D scenes, which allows it to accomplish control tasks involving rigid objects and fluids with significantly better accuracy (Figure 1). Please see our supplementary video for better visualization.

We summarize our contributions as follows: (i) We extend an autoencoding framework with a neural radiance field rendering module and time contrastive learning that allows us to learn 3D-aware scene representations for dynamics modeling and control purely from visual observations. (ii) By incorporating the auto-decoder mechanism at test time, our framework can adjust the learned representation and accomplish the control tasks with the goal specified from camera viewpoints outside the training distribution. (iii) We are the first to augment neural radiance fields using a time-invariant dynamics model, supporting future prediction and novel view synthesis across a wide range of environments with different types of objects.

## 2   Related Work

**3D Scene Representation Learning.** Prior work leverages the latent spaces of autoencoder-like models as learned representations of the underlying 3D scene to enable novel view synthesis from

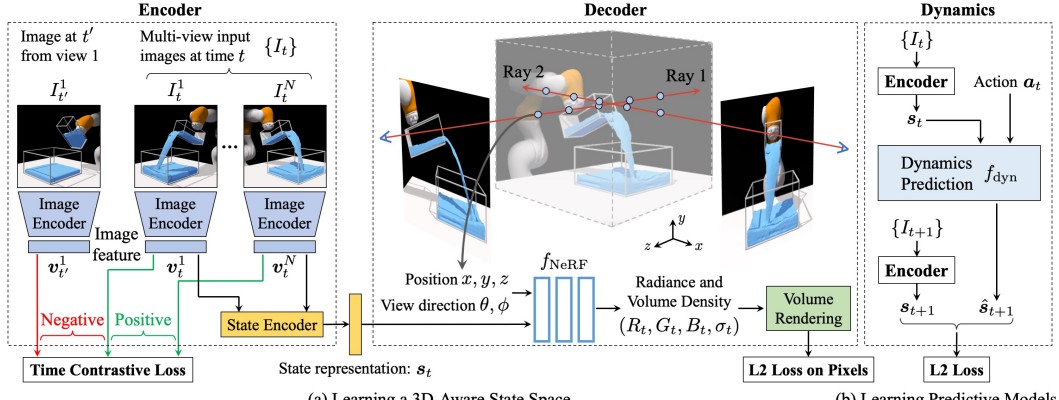

Figure 2: **Overview of the training procedure. Left:** an encoder that maps the input images into a latent scene representation. The images are first sent to an image encoder to generate the image feature representations $v$. Then we combine the image features from the same time step using a state encoder to obtain the state representation $s_t$. A time contrastive loss is applied to enable our model to be invariant to camera viewpoints. **Middle:** a decoder that takes the scene representation as input and generates the visual observation conditioned on a given viewpoint. We use an L2 loss to ensure the reconstructed image to be similar to the ground truth image. **Right:** a dynamics model that predicts the future scene representations $\hat{s}_{t+1}$ by taking in the current representation $s_t$ and action $a_t$. We use an L2 loss to enforce the predicted latent representation to be similar to the scene representation $s_{t+1}$ extracted from the true visual observation $I_{t+1}$.

a single image [24, 25]. Eslami et al. [26] embed this approach in a probabilistic framework. To endow models with 3D structure, voxelgrids can be leveraged as neural scene representations [27, 28, 17, 29, 30], while others have tried to predict particle sets from images [31] or embed an explicit 3D representation to enable inference from never-before-seen viewpoints [32]. Sitzmann et al. [20] propose to learn neural implicit representations of 3D shape and appearance supervised with posed 2D images via a differentiable renderer. Generalizing across neural implicit representations can also be realized by local conditioning on CNN features [33, 34, 35], but this does not learn a global representation of the scene state. Alternatively, gradient-based meta-learning has been proposed for faster inference of implicit neural representations [36]. Deformable scenes can be modeled by transporting input coordinates to neural implicit representations with an implicitly represented flow field or time-variant latent code [37, 38, 39, 40, 41, 42, 43, 44, 45]; however, they typically fit one trajectory and cannot handle different initial conditions and external action inputs, limiting their use in control.

**Model-Based RL in Robotic Manipulation.** We can categorize model-based RL methods by whether they use physics-based or data-driven models, and whether they assume full state access or only visual observation. Methods that rely on physics-based models typically assume full-state information of the environment [46, 47] and require the knowledge of the object models, making them hard to generalize to novel objects or partially observable scenarios. For data-driven models, people have attempted to learn a dynamics model for closed-loop planar pushing [48] or dexterous manipulation [49]. Schenck and Fox [50, 51] tackle a similar fluid pouring task via closed-loop simulation. Although they have achieved impressive results, they rely on state estimators customized for specific tasks, limiting their applicability to more general and diversified manipulation tasks.

Various model-based RL methods have been proposed to learn state representations from visual observations, such as image-space dynamics [4, 5, 1, 7], keypoint representation [52, 9, 12], and low-dimensional latent space [13, 15, 2, 3]. Some works learn a meaningful representation space using reconstruction loss [15, 2]. Others jointly train the forward and inverse dynamics models [14], or use time contrastive loss to regularize the latent embedding [16]. We differ from the previous methods by explicitly incorporating a 3D volumetric rendering process during training.

# 3    3D-Aware Representation Learning for Dynamics Modeling

Inspired by Neural Radiance Fields (NeRF) [19], we propose a framework that learns a viewpoint-invariant model for dynamic environments. As shown in Figure 2, our framework has three parts: (1) an encoder that maps the input images into a latent state representation, (2) a decoder that generates

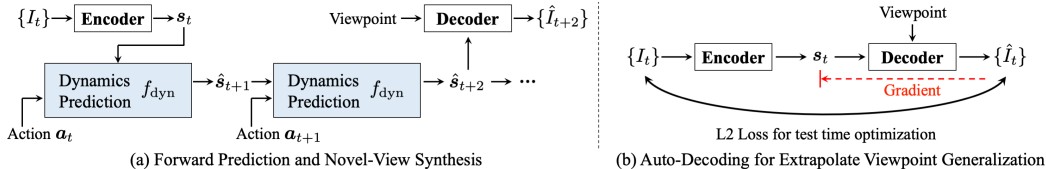

(a) Forward Prediction and Novel-View Synthesis     (b) Auto-Decoding for Extrapolate Viewpoint Generalization

Figure 3: **Forward prediction and viewpoint extrapolation.** (a) We first feed the input image(s) at time $t$ to the encoder to derive the scene representation $s_t$. The dynamics model then takes $s_t$ and the corresponding action sequence as input to iteratively predict the future. The decoder synthesizes the visual observation conditioned on the predicted state representation and an input viewpoint. (b) We propose an auto-decoding inference-via-optimization framework to enable extrapolated viewpoint generalization. Given an input image $I_t$ taken from a viewpoint outside the training distribution, the encoder first predicts the scene representation $s_t$. Then the decoder reconstructs the observation $\hat{I}_t$ from $s_t$ and the camera viewpoint from $I_t$. We calculate the L2 distance between $I_t$ and $\hat{I}_t$ and backpropagate the gradient through the decoder to update $s_t$. The updating process is repeated for $K$ iterations, resulting in a more accurate $s_t$ of the underlying 3D scene.

an observation image under a certain viewpoint based on the state representation, and (3) a dynamics model that predicts the future state representations based on the current state and the input action.

## 3.1 3D-Aware Scene Representation Learning

**Neural Radiance Field.** Given a 3D point $x \in \mathbb{R}^3$ in a scene and a viewing direction unit vector $d \in \mathbb{R}^3$ from a camera, NeRF learns to predict a volumetric radiance field. This is represented using a differentiable rendering function $f_{\text{NeRF}}$ that predicts the corresponding density $\sigma$ and RGB color $c$ using $f_{\text{NeRF}}(x, d) = (\sigma, c)$. To render the color of an image pixel, NeRF integrates the information along the camera ray using $\hat{C}(r) = \int_{h_{\text{near}}}^{h_{\text{far}}} T(h)\sigma(h)c(h)dh$, where $r(h) = o + hd$ is the camera ray with its origin $o \in \mathbb{R}^3$ and unit direction vector $d \in \mathbb{R}^3$, and $T(h) = \exp(-\int_{h_{\text{near}}}^{h} \sigma(s)ds)$ is the accumulated transparency between the pre-defined near depth $h_{\text{near}}$ and far depth $h_{\text{far}}$ along that camera ray. The mean squared error between the reconstructed color $\hat{C}$ and the ground truth $C$ is:

$$\mathcal{L}_{\text{rec}} = \sum_r \|\hat{C}(r) - C(r)\|_2^2. \tag{1}$$

**Neural Radiance Field for Dynamic Scenes.** One key limitation of NeRF is that it assumes the scene is static. For a dynamic scene, it must learn a separate radiance field $f_{\text{NeRF}}$ for each time step. This severely limits the ability of NeRF to be used in planning and control, as it is unable to handle dynamic scenes with different initial configurations or input action sequences. While other models have shown generalization across scenes [20, 53], it's unclear how they can be used in visuomotor control. To enable $f_{\text{NeRF}}$ to model dynamic scenes, we learn an encoding function $f_{\text{enc}}$ that maps the visual observations to a feature representation $s$ for each time step and learn the volumetric radiance field decoding function based on $s$. Let $\{I_t\}$ denotes the set of 2D images that capture a 3D scene at time $t$ from one or more camera viewpoints. The image taken from the $i^{\text{th}}$ viewpoint is represented as $I_t^i$. We use ResNet-18 [54] to extract a feature vector for each image. We take the output of ResNet-18 before the pooling layer and send it to a fully-connected layer, resulting in a 256 dimension image feature $v_t^i$. This image feature is concatenated with the corresponding camera viewpoint information (a 16-D vector obtained by flattening the camera view matrix) and processed using a small multilayer perceptron (MLP) to generate the final image feature. The scene representation $s_t$ at time $t$ is generated by first averaging the image features across multiple camera viewpoints, then being encoded using another small MLP and normalized to have a unit L2 norm.

Given a 3D point $x$, a viewing direction unit vector $d$, and a scene representation $s_t$, we learn a function $f_{\text{dec}}(x, d, s_t) = (\sigma_t, c_t)$ to predict the radiance field represented by the density $\sigma_t$ and RGB color $c_t$. Similar to NeRF, we use the integrated information along the camera ray to render the color of image pixels from an input viewpoint and then compute the image reconstruction loss using Equation 1. During each training iteration, we render two images from different viewpoints to calculate more accurate gradient updates. $f_{\text{dec}}$ depends on the scene representation $s_t$, forcing it to encode the 3D contents of the scene to support rendering from different camera poses.

**Time Contrastive Learning.** To enable the image encoder to be viewpoint invariant, we regularize the feature representation of each image $v_t^i$ using multi-view time contrastive loss (TCN) [16] (see Figure 2a). The TCN loss encourages features of images from different viewpoints at the same time

step to be similar, while repulsing features of images from different time steps to be dissimilar. More specifically, given a time step $t$, we randomly select one image $I_t^i$ as the anchor and extract its image feature $\boldsymbol{v}_t^i$ using the image encoder. Then we randomly select one positive image from the same time step but different camera viewpoint $I_t^{i'}$ and one negative image from a different time step but the same viewpoint $I_{t'}^i$. We use the same image encoder to extract their image features $\boldsymbol{v}_t^{i'}$ and $\boldsymbol{v}_{t'}^i$. Similar to [16], we minimize the following time contrastive loss:

$$\mathcal{L}_{\text{tc}} = \max\left(\|\boldsymbol{v}_t^i - \boldsymbol{v}_t^{i'}\|_2^2 - \|\boldsymbol{v}_t^i - \boldsymbol{v}_{t'}^i\|_2^2 + \alpha, 0\right), \tag{2}$$

where $\alpha$ is a hyper-parameter denoting the *margin* between the positive and negative pairs.

### 3.2 Learning the Predictive Model

After we have obtained the latent state representation $\boldsymbol{s}$, we use supervised learning to estimate the forward dynamics model, $\hat{\boldsymbol{s}}_{t+1} = f_{\text{dyn}}(\boldsymbol{s}_t, \boldsymbol{a}_t)$. Given $\boldsymbol{s}_t$ and a sequence of actions $\{\boldsymbol{a}_t, \boldsymbol{a}_{t+1}, \dots\}$, we predict $H$ steps in the future by iteratively feeding in actions into the one-step forward model. We implement $f_{\text{dyn}}$ as an MLP network which is trained by optimizing the following loss function:

$$\mathcal{L}_{\text{dyn}} = \sum_{h=1}^{H} \|\hat{\boldsymbol{s}}_{t+h} - \boldsymbol{s}_{t+h}\|_2^2, \quad \text{where} \quad \hat{\boldsymbol{s}}_{t+h} = f_{\text{dyn}}(\hat{\boldsymbol{s}}_{t+h-1}, \boldsymbol{a}_{t+h-1}), \quad \hat{\boldsymbol{s}}_t = \boldsymbol{s}_t. \tag{3}$$

We define the final loss as a combination of the image reconstruction loss, the time contrastive loss, and the dynamics prediction loss: $\mathcal{L} = \mathcal{L}_{\text{rec}} + \mathcal{L}_{\text{tc}} + \mathcal{L}_{\text{dyn}}$. We first train the encoder $f_{\text{enc}}$ and decoder $f_{\text{dec}}$ together using stochastic gradient descent (SGD) by minimizing $\mathcal{L}_{\text{rec}}$ and $\mathcal{L}_{\text{tc}}$, which makes sure that the learned scene representation $\boldsymbol{s}$ encodes the 3D contents and is viewpoint-invariant. We then fix the encoder parameters, and train the dynamics model $f_{\text{dyn}}$ by minimizing $\mathcal{L}_{\text{dyn}}$ using SGD. Please see the supplementary materials for the network architecture and training details.

## 4 Visuomotor Control

### 4.1 Online Planning for Closed-Loop Control

When given the goal image $I^{\text{goal}}$ and its associated camera pose, we can feed them through the encoder $f_{\text{enc}}$ to get the state representation for the goal configuration $\boldsymbol{s}^{\text{goal}}$. We use the same method to compute the state representation for the current scene configuration $\boldsymbol{s}_1$. The goal of the online planning problem is to find an action sequence $\boldsymbol{a}_1, \dots, \boldsymbol{a}_{T-1}$ that minimizes the distance between the predicted future representation and the goal representation at time $T$. As shown in Figure 3a, given a sequence of actions, our model can iteratively predict a sequence of latent state representations. The latent-space dynamics model can then be used for downstream closed-loop control tasks via online planning with model-predictive control (MPC). We formally define the online planning problem as follows:

$$\min_{\boldsymbol{a}_1,\dots,\boldsymbol{a}_{T-1}} \|\boldsymbol{s}^{\text{goal}} - \hat{\boldsymbol{s}}_T\|_2^2, \qquad \text{s.t.} \quad \hat{\boldsymbol{s}}_1 = \boldsymbol{s}_1, \hat{\boldsymbol{s}}_{t+1} = f_{\text{dyn}}(\hat{\boldsymbol{s}}_t, \boldsymbol{a}_t). \tag{4}$$

Many existing off-the-shelf model-based RL methods can be used to solve the MPC problem [5, 49, 15, 4, 9, 55, 56]. We experimented with random shooting, gradient-based trajectory optimization, cross-entropy method, and model-predictive path integral (MPPI) planners [57] and found that MPPI performed the best. In our experiments, we specify the action space as the position and orientation of the arm's end-effector. Then, the joint angle of the arm is calculated via inverse kinematics. Please check our supplementary materials for more details on how we solve the planning task.

### 4.2 Auto-Decoder for Viewpoint Extrapolation

End-to-end visuomotor agents can undergo significant performance drop when the test-time visual observations are captured from camera poses outside the training distribution. The convolutional image encoder suffers from the same problem as it is not equivariant to changes in the camera pose, meaning it has a hard time generalizing to out-of-distribution camera views. As shown in Figure 3b, when encountered an image from a viewpoint outside training distribution, with a pixel distribution vastly different from what the model is trained on, passing it through the encoder $f_{\text{enc}}$ will give us an amortized estimation of the scene representation $\boldsymbol{s}_t$. It has a high chance that the decoded image is different from the ground truth as the viewpoint has never been encountered during training.

We fix this problem at test time by applying the inference-by-optimization (also named as an auto-decoding) framework that backpropagates through the volumetric renderer and the neural implicit

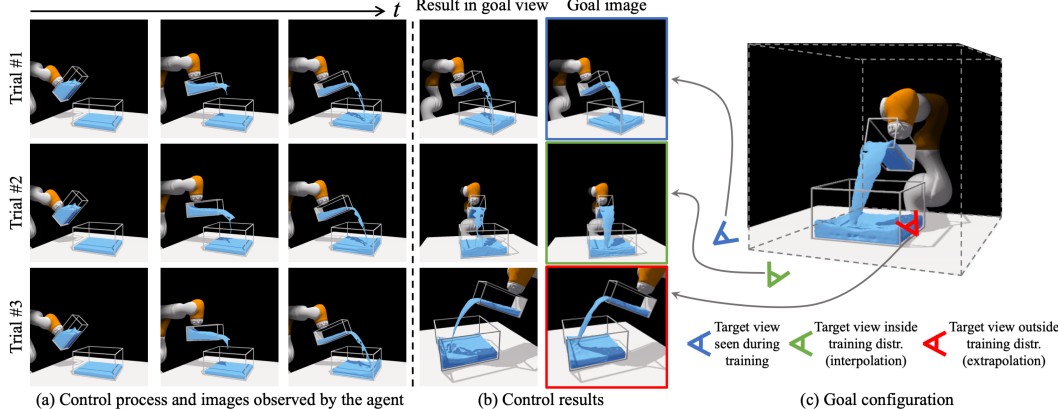

Figure 4: **Qualitative control results of our method on three types of testing scenarios.** The image on the right shows the target configuration we aim to achieve. The left three columns show the control processes, which are also the input images to the agent. The fourth column is the control results from the same viewpoint as the goal image. Trial #1 specifies the goal using a different viewpoint from the agent's but has been encountered during training. Trial #2 uses a goal view that is an interpolation of training viewpoints. Trial #3 uses an extrapolated viewpoint that is outside the training distribution. Our method performs well in all settings.

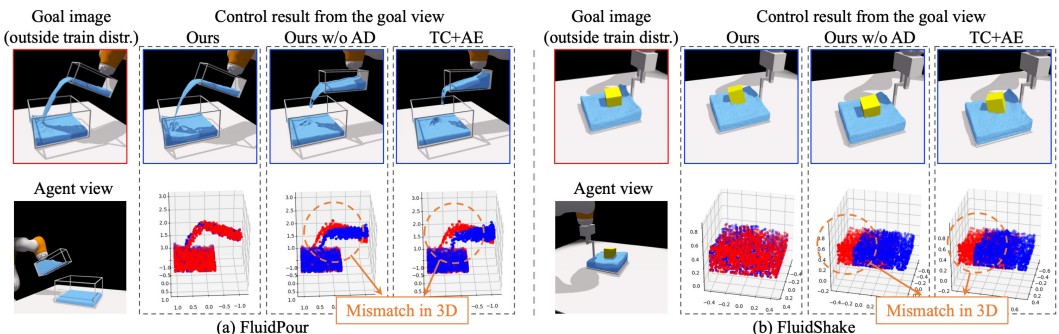

Figure 5: **Qualitative comparisons between our method and baseline approaches on the control tasks.** We show the closed-loop control results on the FluidPour and FluidShake environments. The goal image viewpoint (top-left image of each block) is outside the training distribution and is different from the viewpoint observed by the agent (bottom-left image of each block). Our final control results are much better than a variant that does not perform auto-decoding test-time optimization (Ours w/o AD) and the best-performing baseline (TC+AE), both of which fail to accomplish the task and their control results (blue points) exhibit an apparent deviation from the target configuration (red points) when measured in the 3D points space of the fluids and floating cube.

representation into the state estimate [23, 20]. This is inspired by the fact that the rendering function, $f_{\text{dec}}(\boldsymbol{x}, \boldsymbol{d}, \boldsymbol{s}_t) = (\sigma_t, \boldsymbol{c}_t)$ is viewpoint equivariant, where the output only depends on the state representation $\boldsymbol{s}_t$, the location $\boldsymbol{x}$, and the ray direction $\boldsymbol{d}$, meaning that the output is invariant to the camera position along the camera ray, i.e., even we move the camera closer or farther away along the camera ray, $f_{\text{dec}}$ still tends to generate the same results. We leverage this property and calculate the L2 distance between the input image and the reconstructed image $\mathcal{L}_{\text{ad}} = \|I_t - \hat{I}_t\|_2^2$, and then update the scene representation $\boldsymbol{s}_t$ using stochastic gradient descent. We repeat this updating process $K$ times to derive the state representation of the underlying 3D scene. Note that this update only changes the scene representation while keeping the parameters in the decoder fixed. The resulting representation is used as $\boldsymbol{s}^{\text{goal}}$ in Equation 4 to solve the online planning problem.

## 5 Experiments

**Environments.** We consider the following four environments involving both fluid and rigid objects to evaluate the proposed model and baseline approaches. The environments are simulated using NVIDIA FleX [58]. (1) FluidPour (Figure 7a): This environment contains a fully-actuated cup that pours fluids into a container at the bottom. (2) FluidShake (Figure 7b): A fully-actuated container

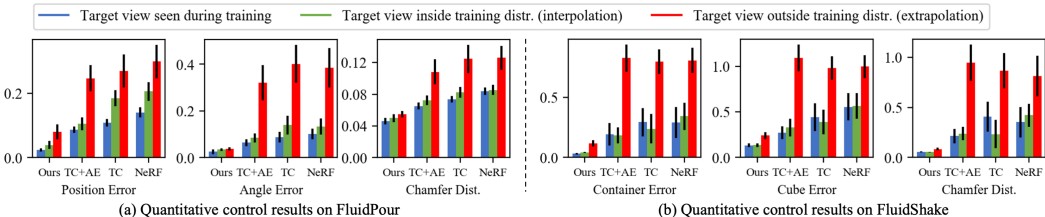

Figure 6: **Quantitative comparisons between our method and baselines on the FluidPour and FluidShake.** In each environment, we compare the results using three different evaluation metrics under three settings, i.e., (1) the target image view seen during training, (2) the target image view is inside the training distribution but not seen during training (interpolation), and (3) the target image view is outside the training distribution (extrapolation). The height of the bars indicates the mean, and the error bar denotes the standard error. Our model significantly outperforms all baselines under all testing settings.

moves on a 2D plane. Inside the container are fluids and a rigid cube floating on the surface. (3) RigidStack (Figure 7c): Three rigid cubes form a vertical stack and are released from a certain height but in different horizontal positions. They fall down and collide with each other and the ground. (4) RigidDrop (Figure 7d): A cube falls down from a certain height. There is a container fixed at a random position on the ground. The cube either falls into the container or bounces out. We use 20 camera views for all environments and generated 1,000 trajectories of 300 time steps for both FluidPour and FluidShake as an offline dataset to train the model, 800 trajectories of 80 time steps for RigidStack, and 1,000 trajectories of 50 time steps for RigidDrop.

**Evaluation Metrics.** We use the first two environments, i.e., FluidPour and FluidShake, to measure the control performance, where we specify the target configuration of the control task using images from (1) one of the viewpoints encountered during training, (2) an interpolated viewpoint between training viewpoints, and (3) an extrapolated viewpoint outside the training distribution (Figure 4).

We provide quantitative evaluations on the control performance in FluidPour and FluidShake by extracting the particle set from the simulator, and measuring the Chamfer distance between the result and the goal, which we denote as "Chamfer Dist.". In FluidPour, we provide additional measurements on the L2 distance of the position/orientation of the cup towards the goal, denoting as "Position Error" and "Angle Error" respectively. In FluidShake, we calculate the L2 distance of the container and cube's position towards the goal and denote them as "Container Error" and "Cube Error" respectively.

## 5.1 Baseline Methods

For comparison, we consider the following three baselines: **TC**: Similar to TCN [16], it only uses time contrastive loss for learning the image feature without having to reconstruct the scene. We learn a dynamics model directly on the image features for control. **TC+AE**: Instead of using Neural Radiance Fields to reconstruct the image, this method uses the standard convolutional decoder to reconstruct the target image when given a new viewpoint. This would then be similar to GQN [26] augmented with a time contrastive loss. **NeRF**: This method is a direct adaptation from the original NeRF paper [19] and is the same as ours except that it does not include the time contrastive loss during training and the auto-decoding test-time optimization. We use the same dynamics model shown in Figure 2b and train the model for each baseline respectively for dynamic prediction. We use the same feedback control method, i.e., MPPI [57], for our model and all the baselines.

## 5.2 Control Results

**Goal Specification from Novel Viewpoints.** Figure 4c shows the goal configuration, and we ask the learned model to perform three control trials where the goal is specified from different types of viewpoints. The left three columns show the MPC control process from the agent's viewpoint. The fourth column visualizes the final configuration the agent achieves from the same viewpoint as the goal image. Trial #1 specifies the goal using a different viewpoint from the agent's but has been encountered during training. Trial #2 uses a goal view that is an interpolation of training viewpoints. Our agent can achieve the target configuration with decent accuracy. For trial #3, we specify the goal view by moving the camera closer, higher, and more downwards with respect to the container. Note this goal image view is outside the distribution of training viewpoints. With the help of test-time

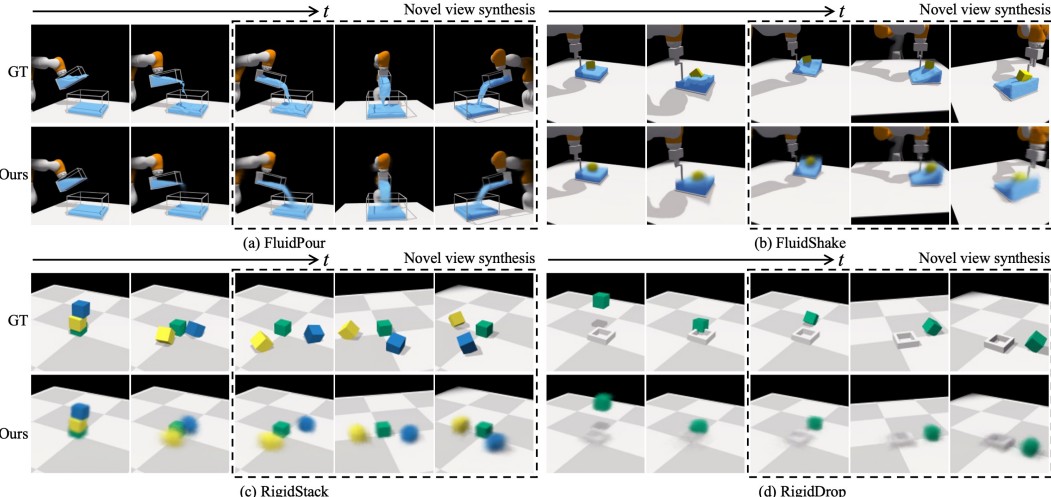

Figure 7: **Forward prediction and novel view synthesis on four environments.** Given a scene representation and an input action sequence, our dynamics model predicts the subsequent latent scene representation, which is used as the input of our decoder model to reconstruct the corresponding visual observation based on different viewpoints. In each block, we render images based on the open-loop future dynamic prediction from left to right. Images in the dotted box compare our model's novel view synthesis results in the last time step with the ground truth from three different viewpoints.

auto-decoding optimization introduced in Section 4.2, our method can successfully achieve the target configuration, as shown in the figure.

**Baseline Comparisons.** We benchmark our model with the baselines by assessing their performance on the downstream control tasks. Figure 5 shows the qualitative comparison between our model (Ours), a variant of our model that does not perform the auto-decoding test-time optimization (Ours w/o AD), and the best-performing baseline (TC+AE) introduced in Section 5.1. We find that when the target view is outside the training distribution and vastly different from the agent view, our full method shows a much better performance in achieving the target configuration. The variant without auto-decoding optimization and TC+AE fail to accomplish the task and exhibit an apparent deviation from the ground truth in the 3D points space of the fluids and floating cube. We also provide quantitative comparisons on the control results. Figure 6 shows the performance in the FluidPour and FluidShake environments. We find our full model significantly outperforms the baseline approaches in both environments under all scenarios and evaluation metrics. The results effectively demonstrate the advantages of the learned 3D-aware scene representations, which contain a more precise encoding of the contents in the 3D environments and are capable of extrapolation viewpoint generalization.

### 5.3 Dynamic Prediction and Novel View Synthesis

Conditioned on a scene representation and an input action sequence, our dynamics model $f_{\mathrm{dyn}}$ can iteratively predict the evolution of the scene representations. Our decoder can then take the predicted state representation and reconstruct the corresponding visual observation from a query viewpoint. Figure 7 shows that our model can accurately predict the future and perform novel view synthesis on four environments involving both fluid and rigid objects, suggesting its usefulness in trajectory optimization. Please check our video results in the supplementary material for better visualization.

## 6 Conclusion

In this paper, we proposed to learn viewpoint-invariant 3D-aware scene representations from visual observations using an autoencoding framework augmented with a neural radiance field rendering module and time contrastive learning. We show that the learned 3D representations perform well on the model-based visuomotor control tasks. When coupled with an auto-decoding test-time optimization mechanism, our method allows goal specification from a viewpoint outside the training distribution. We demonstrate the applicability of the proposed framework in a range of complicated physics environments involving rigid objects and fluids, which we hope can facilitate future works on visuomotor control for complex and dynamic 3D manipulation tasks.

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
