# OpenReview forum: "3D Neural Scene Representations for Visuomotor Control"
_robot-learning.org/CoRL/2021/Conference — CoRL2021 Oral_

### Official Review · Reviewer_1uVz · 2021-07-16

**Originality:** Very Good
**Technical Quality:** Good
**Clarity Of Presentation:** Very Good
**Impact:** 4

**Recommendation:**

Strong Accept: I recommend accepting the paper and will argue for my recommendation even if other reviewers hold a different opinion.

**Summary:**

The authors propose learning a (multiview) latent space dynamics model where the latent state is (self)supervised via both a time contrastive loss and a NERF-based autoencoder. By parameterizing the radiance field with both the latent state and viewpoint, they allow a single NERF to model the dynamic scene as it changes across time. At inference time, the state vector (for the goal state only) is estimated by backpropagating through the NERF decoder (keeping weights frozen) and updating only the state parameters (i.e. an auto-decoding framework similar to DeepSDF). The authors show that view equivariance endemic to NERFs allows for minimal performance decrease using an MPPI planner with novel (out of distribution) goal viewpoints.

**Issues:**

Please see above comment about including more experimental details such as the amount of data used to train for each task and the training time required. The paper would be stronger with a more fully fleshed out appendix addressing these details.

A few other questions:
- I'd also like the authors to clarify exactly what rendered viewpoints were used to train the NERF representation. Is the NERF portion trained with viewpoints similar to the "out of distribution" evaluation views?

- Additionally, why not use the same approach (auto-decoder though the NERF instead of using an encoder) for estimating the current state during control, is it just computationally too expensive to do this at every timestep? This would allow the policy to generalize to different ego perspectives.

- What is the nominal trajectory used with MPPI, is it random is does it come from somewhere (e.g. a demonstration)?

As a side (minor) comment, this related work may also be relevant: the authors embedded an explicit 3D representation into a language-grounding classification task to enable inference from never-before-seen viewpoints). Grounding language attributes to objects using bayesian eigenobjects (Cohen et al. 2019)

**Reviewer Expertise:**

Excellent: Expert knowledge on the topic of the paper

**Strengths And Weaknesses:**

The overall idea of the paper is laudable. It builds on recent advances in 3D scene representations and (factorized) visuomotor control and proposes an elegant and intuitive solution for introducing additional inductive bias in the form of a 3D spatial representation. The experiments suggest that this sort of approach is very well suited to reducing the amount of data required for visuomotor generalization by reducing the need to train across many goal viewpoints.

The biggest weakness of this paper is it's failure to address the practicality of the method in terms of data efficiency, computation, and robustness to real-world scene variation.

On the data efficiency and computation fronts, the paper would be vastly strengthened by more details on the number of training examples required and the compute time. NERFs are notorious for being expensive to fit; how many views (and from what distribution) are used to fit a dynamic scene during training and how long does phase 1 (the portion learning the state representation and NERF) take to train?

Experimentally, this paper's impact is somewhat lessened by its lack of real-world data. Because the simulation environments are consistent, things like lighting and shadows will be exactly the same between training and inference. The current process fits a NERF (and state representation) to a very specific scene setup and it's not clear how this approach will work in the physical world when things like lighting shift and other sources of variation may occur between training and inference. The authors also do not use the auto-decoder procedure on the robot's ego views, only on the goal viewpoint. It would be good to discuss this choice, naively it seems like having ego-view invariance would also be pretty valuable (imagine just setting a camera down anywhere in the robot's workspace instead of keeping it fixed to a single location).


**Summary Of Recommendation:**

While this paper does have shortcomings (I would have liked to see additional experiments in less carefully controlled environments and a discussion of some of the practical aspects of the method) it presents an original and useful contribution to the field. The results presented here are promising and should encourage others in the community to build on this sort of spatially supervised approach.

Update (post Author Response): In light of the thorough responses from the authors and additional information and experiments provided, I'm updating my "Clarity Of Presentation" score from "Good" to "Very Good". I think that this paper is fundamentally novel, interesting, and valuable to the field and should be accepted.

---

> ### Author Response · Authors · 2021-08-31
> **Thank you for your time to review our paper!**
>
> > *On the data efficiency and computation fronts, the paper would be vastly strengthened by more details on the number of training examples required and the compute time.*
>
> The training and environment details have been included in Section D & E in the supplementary PDF in our original submission. Specifically, we use 1,000 trajectories of 300 time steps and 20 camera views for both the FluidPour and FluidShake environments. Using one GPU card, it takes about two days to train the representation model and one day to obtain a good predictive model. We have moved these details to the main paper in the revised version (L218-L221).
>
> > *Experimentally, this paper's impact is somewhat lessened by its lack of real-world data.*
>
> During the rebuttal period, we have conducted experiments on real-world data. As shown in the video (https://anonymous59app.github.io/CoRL-rebuttal/ - Real World Experiments), we use four D415 RGBD cameras to record a human subject pouring water from one cup to another. The cameras are calibrated and synchronized with the frequency of 15 Hz. We recorded 50 pouring episodes of length 15 seconds, resulting in a dataset of ~43,400 frames. We use the first 45 episodes for training and the remaining for testing.
>
> To obtain the action, i.e., the movement of the cup that pours water out, we attached an AprilTag [1] on the cup to obtain the 6 DoF pose of the cup at each time step. From the generated video, we found that our model can make open-loop future predictions on the testing trajectories in the representations space, i.e., given a latent scene representation of the current time step, the subsequent action sequence, our dynamic model can accurately predict the evolution of the latent scene representation and render the corresponding frames that closely resemble the ground truth.
>
> > *Why not use the same approach (auto-decoder though the NERF instead of using an encoder) for estimating the current state during control, is it just computationally too expensive to do this at every timestep? This would allow the policy to generalize to different ego perspectives.*
>
> It is entirely possible to use the auto-decoder on the ego view, which will provide ego-view invariance to out-of-distribution viewpoints. However, as you have also suggested in the review, it is computationally too expensive to do at every timestep; therefore, we did not use it in the final version of our model. For applications that are not bounded by compute time, we could definitely consider applying auto-decoder test-time optimization for the ego view.
>
> > *I'd also like the authors to clarify exactly what rendered viewpoints were used to train the NERF representation. Is the NERF portion trained with viewpoints similar to the "out of distribution" evaluation views?*
>
> Please see Figure 4 of our supplementary material for a comparison between the viewpoints used for training and the viewpoint extrapolation experiments. We generate the extrapolate viewpoints by setting the camera closer, higher, and facing more downwards with respect to the world origin. It is clear from the figure that the images from viewpoints used during training are very different from the ones used for viewpoint extrapolation when measured using pixel difference. Furthermore, when the goal is provided in the extrapolate viewpoints, both baselines, (1) 2D autoencoder and (2) our model without auto-decoding optimization, fail to deliver satisfying control results (Figure 5 in the main paper).
>
> > *What is the nominal trajectory used with MPPI, is it random is does it come from somewhere (e.g. a demonstration)?*
>
> We did not use any demonstrations, and the nominal action trajectory is initialized by commanding the robot’s end effector to stay still at where it was.
>
> > *Related work*
>
> Thank you for the detailed feedback. We have included the related work in the revised version of our paper (Line 79).
>
>
> [1] Edwin Olson, “AprilTag: A robust and flexible visual fiducial system,” ICRA 2011.

---

### Official Review · Reviewer_i7GY · 2021-07-20

**Originality:** Good
**Technical Quality:** Fair
**Clarity Of Presentation:** Very Good
**Impact:** 4

**Recommendation:**

Weak Accept: I recommend accepting the paper, but will not argue for my recommendation if the majority of other reviewers have a different opinion.

**Summary:**

This paper presents a method for viewpoint-equivariant representation learning for visuomotor control. The approach involves learning implicit 3D representations of tabletop scenes with NeRFs and using time-contrastive loss to enforce time consistency. This implicit dynamic scene representation is used as a forward model for model predictive control to achieve specific state-configurations. The viewpoint-equivariant representations allow for specifying target configuration with goal images from arbitrary viewpoints. Experimental results show that the approach is able to achieve desired state-configurations with fluids and rigid-body objects in simulated setups.

**Issues:**

Major:
See comments above on better evaluation tasks.

Minor:
- L232: “similar to [25]” -> “similar to XX approach [25]”
- Related work: there has been some recent progress in learning dynamic/video NeRFs (https://arxiv.org/abs/2103.02597). Although these-=- methods do not condition NeRFs on actions to learn a dynamics model, it might be still good to cite them.
- How many training images are needed per scene? How long does it take to train each scene?
- Can you provide any quantitative evaluations for Section 5.3? Perhaps L2 distances on nearest neighbor search and novel-view synthesis?


**Reviewer Expertise:**

Very good: Comprehensive knowledge of the area

**Strengths And Weaknesses:**

Strengths:
+ This paper proposes an exciting direction for learning scene-dynamics with 3D awareness – a powerful prior for embodied robotics. Most end-to-end learning methods ignore the underlying 3D environment and try to directly learn from pixels. Enforcing an implicit 3D structure could help robots learn reusable skills that aren’t tethered to a specific viewpoint.
+ The subfield of neural scene representation learning has been gaining a lot of traction in the vision and graphics community. This work could potentially lead to interesting directions for learning scene and object representations for robotics applications.
+ Generally, the paper is well written, the methods are easy to understand, and the figures are informative. The supplementary video is also well made and effectively showcases the capabilities of the approach.


Weaknesses:
- One concern is that the evaluation setup is a bit contrived and doesn’t really showcase the full-potential of the 3D dynamic scene representation in a realistic robotics use-case. In the fluid-pouring task, the goal is to achieve the exact fluid state specified in the goal image and the quantitative evaluations measure the Chamfer distance between the fluid particles and the pose error of the cup. In a realistic use-case it’s hard to provide an exact goal-image of the fluid state at a specific time step, and even so it’s unclear if this sort of specificity is necessary in the first place. Alternatively, the evaluation could measure the robot’s ability to complete end-tasks like pouring the liquid upto a certain level – say “20 ml”, or for the fluid-shaking task – make 4 loops around the container with the cube touching each of the four sides. These end-task evaluations could provide better insights into how important 3D-dynamics are for precise model predictive control.
- Another concern is that using NeRF restricts scene representations to mostly tabletop scenes. It’s unclear how neural implicit representations can be scaled to embodied settings like household kitchens or rooms with partial observability. Further, if we are restricted to a tabletop setting, I am not sure why the camera pose would dramatically change in a real-world setup between the input and goal viewpoints other than minor perturbations to the physical setup. One could imagine a user taking a photo of the intended goal from a third-person perspective, but again capturing the state of a fluid at a certain timestep is a very unnatural way of specifying goals. In summary, the paper currently lacks a strong evaluation use-case where “3D awareness” is absolutely necessary for visuomotor control. Showcasing results on tasks that specifically require 3D-reasoning would substantially strengthen the contributions of the paper.
- How can NeRFs handle stochasticity? Motions of granular media and fluids are inherently stochastic in nature. Directly training a predictive model leads to blurry predictions, as evident in the supplementary figures and videos. This limits the approach to short-horizon tasks where the forward model can only be rolled-out for a few steps into the future.


**Summary Of Recommendation:**

This paper proposes an interesting direction for learning implicit dynamic models with 3D awareness. However, the experimental evaluation is missing a strong robotics use-case that highlights the importance of learning 3D-aware implicit dynamics models for model-predictive control.

**Post Rebuttal**
Thank you for the detailed response and new results. I'm still a bit skeptical about learning visual-prediction models for stochastic tasks that involve fluids (even with VAEs). The blurriness of the real-world results partly reaffirms this. But nonetheless, I think the paper poses an interesting direction for using NeRFs in robotics, which could be valuable to future follow-up works. I am keeping my original score of Weak Accept.

---

> ### Author Response · Authors · 2021-08-31
> **Thank you for your time to review our paper! (Part 1 of 2)**
>
> Thank you for the detailed and constructive feedback!
>
> > *Alternatively, the evaluation could measure the robot’s ability to complete end-tasks like pouring the liquid upto a certain level – say “20 ml”*
>
> We appreciate your suggestions on the tasks, which are indeed great candidates! The reason we choose our current tasks, i.e., matching the target visual observation, is because other task specifications would usually require a reward function defined over the underlying state. You could imagine including a decoding function that takes the current scene representation as input and predicts the corresponding reward. The decoding function can then be used within the MPC pipeline to find the action sequence or optimize a policy function. Since this is not the main focus of this work, we would love to tackle your suggested tasks in future work.
>
> > *Robotics use-case*
>
> Related to the previous point, we have included a new qualitative video (https://anonymous59app.github.io/CoRL-rebuttal/ - Ours vs. PID), where the goal is to leave a small amount of fluid in the container at the end of the control episode. Naively matching the container's position and orientation does not work, as shown in the PID baseline that the top container did not pour any fluid into the bottom container. In contrast, our model first learns to pour a certain amount of fluid out and then tilt the container back to match the target configuration, which we hope can justify the necessity of modeling the fluid dynamics and demonstrate some potential practical use-cases of our current setup.
>
> > *It’s unclear how neural implicit representations can be scaled to embodied settings like household kitchens or rooms with partial observability.*
>
> There are works that use neural implicit representations for household environments like convolutional occupancy networks [1], which show impressive reconstruction results in large-scale environments, e.g., a two-floor building. Although not the main focus of this work, we believe neural implicit representations hold the potential to tackle embodied tasks beyond table-top manipulation settings.
>
> > *How can NeRFs handle stochasticity? … This limits the approach to short-horizon tasks where the forward model can only be rolled-out for a few steps into the future.*
>
> Our current formulation does not directly handle stochasticity. Yet, the module for obtaining the scene representations is essentially an autoencoder. You could imagine augmenting it with techniques, e.g., VAE, to make it better suited for stochastic systems.
>
> We would also like to highlight that the open-loop future prediction videos in the supplementary materials show sensible prediction results over a 200 time step interval, which is as long or longer than many state-of-the-art model-based RL frameworks operating on the visual observations, like Deamer V2 [2] or Muzero [3]. In addition, even if there are discrepancies in the long-term rollout, the model can still be useful if short-term predictions are good enough. In some cases, capturing every detail of fluids or granular materials from visual observation is sometimes impossible and maybe unnecessary to accomplish the tasks.
>
> [1] Peng et al., “Convolutional Occupancy Networks,” ECCV 2020.
>
> [2] Hafner et al., “Mastering Atari with Discrete World Models,” ICLR 2021.
>
> [3] Schrittwieser et al., “Mastering Atari, Go, Chess and Shogi by Planning with a Learned Model,” Nature 2020.

---

> > ### Author Response · Authors · 2021-08-31
> > **Thank you for your time to review our paper! (Part 2 of 2)**
> >
> > > *How many training images are needed per scene? How long does it take to train each scene?*
> >
> > The training and environment details have been included in Section D & E in the supplementary PDF in our original submission. Specifically, we use 1,000 trajectories of 300 time steps and 20 camera views for both the FluidPour and FluidShake environments. Using one GPU card, it takes about two days to train the representation model and one day to obtain a good predictive model. We have moved these details to the main paper in the revised version.
> >
> > > *Can you provide any quantitative evaluations for Section 5.3? Perhaps L2 distances on nearest neighbor search and novel-view synthesis?*
> >
> > We conducted additional experiments measuring the accuracy in finding the nearest neighbors (NN) across viewpoints, and the following table shows the results.
> >
> > | env | Ours | NN in pixel space | Random |
> > | ----------- | ----------- | ----------- | ----------- |
> > | FluidPour | 1.772718 | 147.971993 | 99.903261 |
> > | FluidShake | 2.584928 | 132.467998 | 99.646617 |
> >
> > Specifically, for each query frame, the model is asked to find the closest frame from a randomly selected viewpoint from a trajectory with 300 frames. We measure the accuracy using the L1 distance between the time indexes. Randomly selecting the closest frame leads to an averaged distance of 100 times steps between the selected frame and the ground truth frame. Selecting based on the pixel difference performs even worse. Our method can accurately find the nearest neighbor. The average time step difference between the selected frame and the ground truth frame is 1.772718 in FluidPour and 2.584928 in FluidShake.
> >
> > > *Model name and related work*
> >
> > Thank you for the detailed feedback. We have corrected the model name and included the related work in the revised version of our paper (Line 233 & 237 and Line 86).

---

### Official Review · Reviewer_5NEJ · 2021-07-24

**Originality:** Excellent
**Technical Quality:** Excellent
**Clarity Of Presentation:** Excellent
**Impact:** 4

**Recommendation:**

Strong Accept: I recommend accepting the paper and will argue for my recommendation even if other reviewers hold a different opinion.

**Summary:**

This paper demonstrates an approach for modelling dynamics of a scene using an implicit neural representation similar to NeRF. The model is based on an autoencoder framework, where an encoder maps first-person images to a latent state representation that encodes the system state, and a decoder takes the state representation and a viewpoint as input and renders an image. A dynamics model in the latent state space enables predicting future states, and applying the decoder to render future observations from arbitrary viewpoints.

At test-time, the system takes a viewpoint and a goal image as input, and uses an optimization process to construct the corresponding scene representation for the goal. This optimization process works better on out-of-training-distribution viewpoints than directly encoding the goal image with the encoder.

The latent state dynamics are learned with a time-contrastive loss that pulls state representations from different viewpoints at the same time-step closer, while pushing representations from the same viewpoint but different time-steps apart. As a result, the latent state vector focuses on the state of the dynamic system in a viewpoint-invariant way, while the decoder is responsible for modelling viewpoint-variant rendering. This makes the state representations amenable for planning and control.

The paper applies MPC to reach a desired goal state represented by an image from an arbitrary viewpoint. The control experiments include two scenes: fluid pouring and moving a water-filled box to create waves. Additionally, the paper shows that the system is capable of novel view synthesis of a collapsing tower of blocks, and a falling cube.


**Issues:**

The main paper would benefit from at least a few sentences commenting on how many viewpoint images, from how many timesteps were needed during training, and how long did the training take per scene. The NeRF approach required a lot of viewpoints and a very expensive optimization. I wonder how that is affected by adding the temporal dimension and the time-contrastive loss.

**Reviewer Expertise:**

Good: General knowledge of the area

**Strengths And Weaknesses:**

Strengths:
- Technically strong and well thought out approach.
- Demonstrates an exciting new result modelling scene dynamics over a latent space on which an implicit volumetric scene representation is conditioned.
- Able to model dynamics of rigid bodies, fluids, and soft bodies.
- Convincing experimental results.

Weaknesses:
- Does not yet enable generalization of dynamics understanding to new scenes.

**Summary Of Recommendation:**

Implicit representations of geometry and rendering have recently seen explosive interest in computer vision and graphics research, due to enabling previously unimaginable results, such as neural rendering from arbitrary viewpoints, scene composition, and 3D reconstructions from still images. Such reasoning about geometry and dynamics could greatly benefit robotics applications as well. Traditionally, model-based reasoning (e.g. MPC, task and motion planning) in 3D has required either hand-specified state representations and dynamics models, running simulators in the loop, or learned models that reason in 2D from fixed viewpoints. If we could instead learn such dynamics models and scene representations in 3D just from observations, that could pave the way for new planning and learning algorithms and breakthroughs. This paper takes a step in that direction and should appear at the conference. The main weakness is the significant amount of learning required for each scene individually and the lack of generalization to new scenes, however this is not unlike most RL research, and also inline with the state of the art in implicit scene representations in vision.

---

> ### Author Response · Authors · 2021-08-31
> **Thank you for your time to review our paper!**
>
> Thank you for the detailed and constructive feedback!
>
> > *Generalization of dynamics understanding to new scenes.*
>
> Our model uses a single latent vector as the scene representation, which is expected to work in dynamical systems well-supported by the training data. If the goal is to generalize to scenes with contents or combinations of contents that the model has never seen before, more structured intermediate representations may be needed to capture the underlying scene structure, which we leave as future work.
>
> > *How many viewpoint images, from how many timesteps were needed during training, and how long did the training take per scene.*
>
> The training and environment details have been included in Section D & E in the supplementary PDF in our original submission. Specifically, we use 1,000 trajectories of 300 time steps and 20 camera views for both the FluidPour and FluidShake environments. Using one GPU card, it takes about two days to train the representation model and one day to obtain a good predictive model. We have moved these details to the main paper in the revised version (L218-L221).

---

### Official Review · Reviewer_BFGN · 2021-07-25

**Originality:** Very Good
**Technical Quality:** Good
**Clarity Of Presentation:** Good
**Impact:** 4

**Recommendation:**

Weak Accept: I recommend accepting the paper, but will not argue for my recommendation if the majority of other reviewers have a different opinion.

**Summary:**

This paper claims it is possible to learn a representation of a scene, that can be used both to predict new views and future views. It demonstrates this claim on several simple scenes, including some involving particle systems.


**Issues:**

See the weaknesses above to make the paper more convincing.


**Reviewer Expertise:**

Good: General knowledge of the area

**Strengths And Weaknesses:**

Strengths

This is an exciting claim.

The paper shows it can be done on some simple examples, which will inspire future work.

Weaknesses.

These are not real images/videos or a real robot. It is claimed Covid made this difficult, although very simple and cheap ($2K) robots could be installed at home to do the evaluation with.

"Great claims require great evidence". It is worth looking skeptically at the evaluation tasks.

1) All of them involved simple clean images without clutter (much less other things moving in the scenes). It appears that the system knew
to
only look at the blue particle systems (based on Figure 5 lower row) or the cubes, and not the containers or the robot or table.
Real pouring of transparent liquids are hard to capture.
So I think we have to be skeptical about the claims about learning vision.

2) It appears that the state vector for dynamics for these systems is pretty simple (positions and velocities of the robot tip or block release
points. The space that needs to be searched is relatively simple. To what extent is the system just matching the
motion of the robot?

3) It appears the time intervals are short.

4) It does not appear the physics for the "liquid" are realistic.
There is something physically wrong with the pours in Figures 4
and 5: The arc of the pour is too smooth and goes out too far. There is
little to provide the large horizontal velocity needed
to the liquid (or particles). The pour should go almost straight down,
given the container is not moving horizontally at a high rate during
the time interval.
I  suspect the amount
of pour and sloshing is not done well either.

5) Figure 5 shows that the major thing going on is overlapping in 3D the
blue stuff by translating the containers. It is probably the same for the cubes. If the width of the
pour is small, the system will probably be happy with no pour. There is
no notion of liquid levels changing over time forcing pouring to occur.

In FluidShake is the position of the yellow cube being controlled?

Usually systems like these are flooded with irrelevant
features, and the type of learning used can't figure out what is relevant or irrelevant with a reasonable amount of training data.

I am skeptical that this system can handle color/texture map variations on the objects, or changing the shape or size of the containers
or blocks, or even movement of the table.



**Summary Of Recommendation:**

This paper makes a claim and some progress on behavior capture for liquid/granular materials, and, although imperfect, should be accepted.

---

> ### Author Response · Authors · 2021-08-31
> **Thank you for your time to review our paper!**
>
> Thank you for the detailed and constructive feedback!
>
> > *These are not real images/videos or a real robot.*
>
> During the rebuttal period, we have conducted experiments on real-world data. As shown in the video (https://anonymous59app.github.io/CoRL-rebuttal/ - Real World Experiments), we use four D415 RGBD cameras to record a human subject pouring water from one cup to another. The cameras are calibrated and synchronized with the frequency of 15 Hz. We recorded 50 pouring episodes of length 15 seconds, resulting in a dataset of ~43,400 frames. We use the first 45 episodes for training and the remaining for testing.
>
> To obtain the action, i.e., the movement of the cup that pours water out, we attached an AprilTag [1] on the cup to obtain the 6 DoF pose of the cup at each time step. From the generated video, we found that our model can make open-loop future predictions on the testing trajectories in the representations space, i.e., given a latent scene representation of the current time step, the subsequent action sequence, our dynamic model can accurately predict the evolution of the latent scene representation and render the corresponding frames that closely resemble the ground truth.
>
> > *It appears that the system knew to only look at the blue particle systems (based on Figure 5 lower row) or the cubes, and not the containers or the robot or table.*
>
> We would like to clarify that our model looks at the scene as a whole, including the robot/containers/table. The second row of Figure 5 is only to highlight the comparison between the resulting fluid shape and the fluid shape in the target configuration. We have included the quantitative comparison regarding the control accuracy of the container’s position/orientation in Figure 6.
>
> > *To what extent is the system just matching the motion of the robot? … There is no notion of liquid levels changing over time forcing pouring to occur.*
>
> To show the necessity of modeling the fluid dynamics in our task, we include an additional baseline that uses PID control to reach the robot state in the target image without worrying about the fluid. Note that this baseline uses additional information, including the ground truth state of the robot in both the current and the target image.
>
> The qualitative video (https://anonymous59app.github.io/CoRL-rebuttal/ - Ours vs. PID) shows a controlling trial where the goal is to leave a small amount of fluid in the container at the end of the control episode. Naively matching the container's position and orientation does not work, as shown in the PID baseline that the top container did not pour any fluid into the bottom container. In contrast, our model first learns to pour a certain amount of fluid out and then tilt the container back to match the target configuration.
>
> We further use the Chamfer distance to measure the models' performance in matching the fluid state, where our method significantly outperforms the PID baseline (Ours: 0.048237 vs. PID: 0.085727).
>
> > *It appears the time intervals are short.*
>
> We would like to highlight that the open-loop future prediction videos in the supplementary materials show sensible prediction results over a 200 time step interval, which is as long as or longer than many state-of-the-art model-based RL frameworks operating on the visual observations, like Deamer V2 [2] or Muzero [3].
>
> > *In FluidShake is the position of the yellow cube being controlled?*
>
> Yes, the goal of our model is to match the overall configuration of the target scene. We have also included quantitative evaluation of the cube’s control accuracy in Figure 6b.
>
> > *Usually systems like these are flooded with irrelevant features, and the type of learning used can't figure out what is relevant or irrelevant with a reasonable amount of training data.*
>
> Disentangling the task-relevant and -irrelevant features from the visual observations is an important and currently an active research topic. The focus of this paper is to approach the control tasks by learning 3D-aware scene representations. We are glad to investigate the disentanglement ability in future works.
>
>
> [1] Edwin Olson, “AprilTag: A robust and flexible visual fiducial system,” ICRA 2011.
>
> [2] Hafner et al., “Mastering Atari with Discrete World Models,” ICLR 2021.
>
> [3] Schrittwieser et al., “Mastering Atari, Go, Chess and Shogi by Planning with a Learned Model,” Nature 2020.

---

> > ### Comment · Reviewer_BFGN · 2021-09-02
> > **Advertise new results in actual paper.**
> >
> > I would suggest teasing the new results that were added to the supplementary document in the actual paper. Most people ignore supplementary documents, or never even know they exist somewhere.
> >
> > I also suggest the authors make sure that the paper is updated to take into account all the reviewer comments and your rebuttals. If the reviewers didn't get something or had a question, it is likely other readers will as well. Everything should result in a change to the paper,
> > rather than a comment in a rebuttal to a reviewer which most readers will never see.
> >
> > I have to admit I am skeptical of the real robot results, because the "pour" or liquid moving from one cup to another is hard to see, and does not look continuous from one cup to another. The "results" video is quite blurred, and I could just as well argue it is a great success or that it is a total failure. The pour is so blurred it is hard to tell. It is true the levels in the cups change, but is that connected in any way to the amount of predicted pour (if that is even measurable), and consistant with the change in level in the other cup? Perhaps showing the video twice, once augmented with some kind of overlay that indicates what is being predicted by the system might clarify these issues.

---

> ### Comment · Reviewer_BFGN · 2021-09-02
> **Still think paper should be accepted.**
>
> After the rebuttal process, I still think the paper should be accepted. I defer to the greater wisdom of the other reviewers, area chairs, etc.

---

### Author Response · Authors · 2021-08-31
**Summary of Changes**

We thank the reviewers and the area chair for the thorough and thoughtful comments. We are glad that they found our paper “well written,” the work “well motivated,” and the contribution “technically strong.” The area chair further pointed out that the proposed method “addresses a very important and timely problem within the robotics community” and progresses in this direction “can have a major impact.”

We have provided detailed responses to each reviewer, and in the following, we summarize the changes we have made to the manuscript and the additional experiments. All changes to the manuscript are highlighted in yellow for easier reference.

1. We moved the description of the environment details from the supplementary to the main paper (Section 5 - Environments, line 218-221).
2. We included the suggested references in Section 2 of the main paper (line 79 & 86) and corrected the reference to model names (line 233 & 237).
3. As suggested by the reviewers and area chair, we conducted three additional experiments and included detailed descriptions in the supplementary materials (Section G) and copied them below.
4. We showcase our additional qualitative results at the webpage URL https://anonymous59app.github.io/CoRL-rebuttal/
5. We included a paragraph discussing the limitation and future works in the supplementary materials (Section H).

---

> **Additional experiments using real-world data (Area Chair 9X8h, Reviewer BFGN, i7GY, 1uVz).**

During the rebuttal period, we have conducted experiments on real-world data. As shown in the video (https://anonymous59app.github.io/CoRL-rebuttal/ - Real World Experiments), we use four D415 RGBD cameras to record a human subject pouring water from one cup to another. The cameras are calibrated and synchronized with the frequency of 15 Hz. We recorded 50 pouring episodes of length 15 seconds, resulting in a dataset of ~43,400 frames. We use the first 45 episodes for training and the remaining for testing.

To obtain the action, i.e., the movement of the cup that pours water out, we attached an AprilTag [1] on the cup to obtain the 6 DoF pose of the cup at each time step. From the generated video, we found that our model can make open-loop future predictions on the testing trajectories in the representations space, i.e., given a latent scene representation of the current time step, the subsequent action sequence, our dynamic model can accurately predict the evolution of the latent scene representation and render the corresponding frames that closely resemble the ground truth.

> **Additional comparison with a PID baseline that only matches the robot’s state (Reviewer BFGN).**

To show the necessity of modeling the fluid dynamics in our task, we include an additional baseline that uses PID control to reach the robot state in the target image without worrying about the fluid. Note that this baseline uses additional information, including the ground truth state of the robot in both the current and the target image.

The qualitative video (https://anonymous59app.github.io/CoRL-rebuttal/ - Ours vs. PID) shows a controlling trial where the goal is to leave a small amount of fluid in the container at the end of the control episode. Naively matching the container's position and orientation does not work, as shown in the PID baseline that the top container did not pour any fluid into the bottom container. In contrast, our model first learns to pour a certain amount of fluid out and then tilt the container back to match the target configuration.

We further use the Chamfer distance to measure the models' performance in matching the fluid state, where our method significantly outperforms the PID baseline (Ours: 0.048237 vs. PID: 0.085727).

> **Quantitative evaluations on nearest neighbor search (Reviewer i7GY).**

We conducted additional experiments measuring the accuracy in finding the nearest neighbors (NN) across viewpoints, and the following table shows the results.

| env | Ours | NN in pixel space | Random |
| ----------- | ----------- | ----------- | ----------- |
| FluidPour | 1.772718 | 147.971993 | 99.903261 |
| FluidShake | 2.584928 | 132.467998 | 99.646617 |

Specifically, for each query frame, the model is asked to find the closest frame from a randomly selected viewpoint from a trajectory with 300 frames. We measure the accuracy using the L1 distance between the time indexes. Randomly selecting the closest frame leads to an averaged distance of 100 times steps between the selected frame and the ground truth frame. Selecting based on the pixel difference performs even worse. Our method can accurately find the nearest neighbor. The average time step difference between the selected frame and the ground truth frame is 1.772718 in FluidPour and 2.584928 in FluidShake.

---

### Meta-Review · Area_Chair_9X8h · 2021-08-13

**Recommendation:** Accept (Oral)
**Confidence:** 4

**Metareview:**

**Update after rebuttal**
The authors have addressed most issues raised by the reviewers, and overall I recommend accept.
Two more comments:
* I agree with the reviewer that raised the concern of integrating the results presented in the rebuttal into the paper. Please try to do so as best as you can.
* I still would like to see a more "balanced" discussion on what the limitations are

**Initial meta-review**:
**Summary**:
This work presents an approach towards learning viewpoint-equivariant visual latent representations that can be used to used for visual planning/control (via a learned dynamics model in the latent space). The proposed framework/algorithm allows to specify targets/goals from viewpoints different from the robots viewpoint, and can support goal specification from camera viewpoints that are outside the training distribution.

**Strengths**:
The manuscript addresses a very important and timely problem within the robotics community. Progress on learning visual latent representations and dynamics models that generalize across viewpoints can have major impact.

The manuscript is well written, the work will motivated, and a technically strong contribution is made.

**Weaknesses**:
The reviewers raise some questions that should be addressed by the authors

* Overall, several reviewers raise the issue of evaluation on simulated data only, where the data is very “clean/simple”. It is unclear how well this approach would perform on real world data? The authors should consider performing a subset of the evaluation on real world data (maybe not the downstream task evaluation, but only the learning representations part). Alternatively, there may be be way to introduce some noise?
* it is unclear how much training data from how many viewpoints is required to train the latent representations and predictive model, and how much time it took to train those representations.
* The manuscript is missing a clear discussion of the limitations in terms of generalization and what has been learned. While it is useful to use the big vision/claims for motivation, it is equally important to clearly state what has been achieved with this work, and where the limitations lie. For instance it seems unlikely that the system for the pouring task learns anything about the water levels (eg half full) for various sized containers. A thorough discussion would better enable future research on this topic.

---

> ### Author Response · Authors · 2021-08-31
> **Thank you for your time to provide the meta-review!**
>
> We thank the reviewers for the constructive comments and the area chair for the summarization.
>
> > *It is unclear how well this approach would perform on real world data?*
>
> During the rebuttal period, we have conducted experiments on real-world data. As shown in the video (https://anonymous59app.github.io/CoRL-rebuttal/ - Real World Experiments), we use four D415 RGBD cameras to record a human subject pouring water from one cup to another. The cameras are calibrated and synchronized with the frequency of 15 Hz. We recorded 50 pouring episodes of length 15 seconds, resulting in a dataset of ~43,400 frames. We use the first 45 episodes for training and the remaining for testing.
>
> To obtain the action, i.e., the movement of the cup that pours water out, we attached an AprilTag [1] on the cup to obtain the 6 DoF pose of the cup at each time step. From the generated video, we found that our model can make open-loop future predictions on the testing trajectories in the representations space, i.e., given a latent scene representation of the current time step, the subsequent action sequence, our dynamic model can accurately predict the evolution of the latent scene representation and render the corresponding frames that closely resemble the ground truth.
>
> > *It is unclear how much training data from how many viewpoints is required to train the latent representations and predictive model, and how much time it took to train those representations.*
>
> The training and environment details have been included in Section D & E in the supplementary PDF in our original submission. Specifically, we use 1,000 trajectories of 300 time steps and 20 camera views for both the FluidPour and FluidShake environments. Using one GPU card, it takes about two days to train the representation model and one day to obtain a good predictive model. We have moved these details to the main paper in the revised version (L218-L221).
>
> > *The manuscript is missing a clear discussion of the limitations in terms of generalization and what has been learned.*
>
> The main focus of this work is to obtain 3D-aware representations for visuomotor control, where we only assume to have visual observations of the task environment. We show through experiments that the learned representations more precisely encode the underlying 3D contents than the 2D counterparts, and the inductive bias provided by the neural implicit representation also allows for out-of-distribution viewpoint generalization.
>
> Similar to many other data-driven methods, our model can deliver reasonable performance in regions well-supported by the data. For cases that our model has never seen before, e.g., containers with out-of-distribution shape/size, we wouldn't expect our model to generalize. Potential solutions may include (1) adding the examples and increasing the diversity of the training set or (2) using a more structured/compositional representation space instead of a single vector as in our model, which we leave for future works. We have included the discussion in Section H of the revised supplementary material.
>
> [1] Edwin Olson, “AprilTag: A robust and flexible visual fiducial system,” ICRA 2011.

---

### Decision · Program_Chairs · 2021-09-13

**Decision:**

Accept (Oral)

**Comment:**

**Update after rebuttal**
The authors have addressed most issues raised by the reviewers, and overall I recommend accept.
Two more comments:
* I agree with the reviewer that raised the concern of integrating the results presented in the rebuttal into the paper. Please try to do so as best as you can.
* I still would like to see a more "balanced" discussion on what the limitations are

**Initial meta-review**:
**Summary**:
This work presents an approach towards learning viewpoint-equivariant visual latent representations that can be used to used for visual planning/control (via a learned dynamics model in the latent space). The proposed framework/algorithm allows to specify targets/goals from viewpoints different from the robots viewpoint, and can support goal specification from camera viewpoints that are outside the training distribution.

**Strengths**:
The manuscript addresses a very important and timely problem within the robotics community. Progress on learning visual latent representations and dynamics models that generalize across viewpoints can have major impact.

The manuscript is well written, the work will motivated, and a technically strong contribution is made.

**Weaknesses**:
The reviewers raise some questions that should be addressed by the authors

* Overall, several reviewers raise the issue of evaluation on simulated data only, where the data is very “clean/simple”. It is unclear how well this approach would perform on real world data? The authors should consider performing a subset of the evaluation on real world data (maybe not the downstream task evaluation, but only the learning representations part). Alternatively, there may be be way to introduce some noise?
* it is unclear how much training data from how many viewpoints is required to train the latent representations and predictive model, and how much time it took to train those representations.
* The manuscript is missing a clear discussion of the limitations in terms of generalization and what has been learned. While it is useful to use the big vision/claims for motivation, it is equally important to clearly state what has been achieved with this work, and where the limitations lie. For instance it seems unlikely that the system for the pouring task learns anything about the water levels (eg half full) for various sized containers. A thorough discussion would better enable future research on this topic.